# QTL Mapping and Developing KASP Markers for High-Temperature Adult-Plant Resistance to Stripe Rust in Argentinian Spring Wheat William Som (PI 184597)

**DOI:** 10.3390/ijms26115072

**Published:** 2025-05-24

**Authors:** Arjun Upadhaya, Meinan Wang, Chao Xiang, Nosheen Fatima, Sheri Rynearson, Travis Ruff, Deven R. See, Michael Pumphrey, Xianming Chen

**Affiliations:** 1Department of Plant Pathology, Washington State University, Pullman, WA 99164, USA; arjun.upadhaya@wsu.edu (A.U.); meinan_wang@wsu.edu (M.W.); xiangchao@scsaas.cn (C.X.); nosheen.fatima@wsu.edu (N.F.); travis.ruff@usda.gov (T.R.); deven.see@usda.gov (D.R.S.); 2Crop Research Institute, Sichuan Academy of Agricultural Sciences and Environment-Friendly Crop Germplasm Innovation and Genetic Improvement Key Laboratory of Sichuan Province, Chengdu 610066, China; 3Department of Crop and Soil Sciences, Washington State University, Pullman, WA 99164, USA; sbabb@wsu.edu (S.R.); m.pumphrey@wsu.edu (M.P.); 4USDA-ARS Wheat Health, Genetics, and Quality Research Unit, Pullman, WA 99164, USA

**Keywords:** high-temperature and adult plant resistance, KASP markers, quantitative trait loci, stripe rust, wheat

## Abstract

Stripe rust, caused by *Puccinia striiformis* f. sp. *tritici* (*Pst*), is a destructive disease of wheat worldwide. William Som (WS), an Argentinian spring wheat landrace, has consistently exhibited high-level resistance to stripe rust for over 20 years in our field evaluations in Washington state, USA. A previous study showed high-temperature adult-plant (HTAP) resistance in WS. To map the HTAP resistance quantitative trait loci (QTL) in WS, 114 F_5-8_ recombinant inbred lines (RILs) from the cross AvS/WS were evaluated for their stripe rust response in seven field environments in Washington. The RILs and parents were genotyped with the Infinium 90K SNP chip. Four stable QTL, *QYrWS.wgp-1BL* on chromosome 1B (669–682 Mb), *QyrWS.wgp-2AL* on 2A (611–684 Mb), *QyrWS.wgp-3AS* on 3A (9–13 Mb), and *QyrWS.wgp-3BL* on 3B (476–535 Mb), were identified, and they explained 10.0–19.0%, 10.2–16.7%, 7.0–15.9%, and 12.0–27.8% of the phenotypic variation, respectively. The resistance in WS was found to be due to additive interactions of the four QTL. For each QTL, two Kompetitive allele-specific PCR (KASP) markers were developed, and these markers should facilitate the introgression of the HTAP resistance QTL into new wheat cultivars.

## 1. Introduction

Stripe rust, or yellow rust, caused by *Puccinia striiformis* f. sp. *tritici* (*Pst*), is one of the most significant biotic constraints affecting wheat production worldwide [1,2,3,4]. The disease can cause up to 100% yield losses in susceptible cultivars under conducive conditions. Since the first detection of stripe rust in North America in 1915, numerous epidemics have been documented on the continent, some localized and others on a national or continental scale [5,6,7]. The most notorious stripe rust epidemics in North America were those from the United States (US) Pacific Northwest (PNW) region between 1960 and 1964 [1,7]. During this four-year period, the US PNW region (Washington, Oregon, Idaho, and Montana) suffered substantial economic losses. For instance, in Washington state alone, the estimated losses were $15 million in 1960 and $30 million in 1961 [1,7]. Following these epidemics, the deployment of stripe rust resistance genes has become one of the top priorities for breeding programs in the region, and consequently cultivars with varying degrees of resistance have been released to keep stripe rust under control [6,7,8,9]. However, due to the production of both spring and winter wheat crops and susceptible grass species for the constant presence of host plants; the favorable weather conditions for the pathogen to infect, grow, and survive; and its ability to quickly evolve new virulence genes and combinations against widely deployed resistance genes, the PNW wheat production remains under constant *Pst* threat [6,7,10]. Since 2000, stripe rust has changed from a disease mainly in the western states to a major problem throughout the US [1,10,11]. Whenever a stripe rust epidemic occurs in the Great Plains, the damage is huge. For example, the yield losses were estimated at 25% in Oklahoma and 15% in Kansas, and the countrywide yield loss reached 8.7% or over 4.5 million metric tons [1].

Stripe rust can be effectively managed by growing resistant cultivars or applying fungicides [1,6,7,12]. The use of chemical fungicides incurs additional costs to growers and is hazardous to the environment. Therefore, growing resistant cultivars is the best approach for sustainable management of the disease. There are two major types of resistance to stripe rust, all-stage resistance (ASR) and high-temperature adult-plant (HTAP) resistance [6,7,12]. ASR, also called seedling resistance, is expressed throughout all plant growth stages and can be detected early at the seedling stage. While ASR genes generally provide higher levels of resistance, they are effective against only avirulent races and tend to be overcome by virulent races within a few years of deployment in cultivars [6,10]. Thus, ASR genes alone may not provide durable resistance to stripe rust.

In contrast, HTAP resistance is non-race-specific and is, therefore, durable [6,7,8,13,14]. HTAP resistance can be controlled by one or more genes and is expressed as plants grow old and temperatures are high [6,7,13]. The expression of HTAP resistance usually begins at the stem elongation stage, then gradually increasing, and it becomes prominent on flag leaves. The typical resistant phenotype associated with HTAP resistance includes necrotic stripes with or without uredinia [6]. The level of HTAP resistance varies from gene to gene, depending upon the number of genes and gene combinations, and influenced by temperatures and plant growth stages. To confirm the presence of HTAP resistance in a genotype, both seedlings and adult plants should be tested with the same *Pst* races under controlled greenhouse conditions with low and high temperature profiles [6]. HTAP resistance has been widely used for over 60 years in wheat cultivars grown in the US PNW region [6,7,8,13,14]. As HTAP resistance is quantitative and influenced by the stripe rust pressure, plant growth stage, and temperature, the best approach is combining ASR and HTAP resistance genes to take advantage and overcome the disadvantages of both types of resistance [6]. It is essential to identify more genes for both ASR and HTAP types of resistance to stripe rust.

To date, 87 *Yr* (Yellow rust resistance) genes have been permanently designated for resistance to stripe rust in wheat [2,12,15], of which *Yr18*, *Yr29*, *Yr36*, *Yr39*, *Yr46*, *Yr48*, *Yr49*, *Yr52*, *Yr54*, *Yr55*, *Yr58*, *Yr59*, *Yr62*, *Yr71*, *Yr75*, *Yr78*, and *Yr79* confer HTAP resistance [12,16,17,18,19,20,21,22,23,24]. In addition to these named *Yr* genes, quantitative trait loci (QTL) conferring HTAP resistance have been identified in wheat landraces, breeding materials, and cultivars [6,9,18,25,26,27,28], including *QYrex.wgp-1BL*, *QYrex.wgp-3BL*, and *QYrex.wgp-6AS* in Express [18]; *QYrlu.cau-2BS1* and *QYrlu.cau-2BS2* in Luke [27]; *QYrMa.wgp-3BS* and *QYrMa.wgp-6BS* in Madsen [25]; and *QYrst.wgp-6BS.1* and *QYrst.wgp-6BS.2* in Stephens [26]. The PNW wheat cultivars Alpowa, Luke, Madsen, and Stephens, which were released in the last quarter of the 20th century and once widely popular cultivars in this region, have continued serving as genetic stocks for HTAP resistance in wheat breeding programs in this region and other regions. More genetic stocks, genes, and useful markers are still needed to enrich the stripe rust resistance sources in breeding programs.

The advancements in sequencing technologies have enabled the identification of genome-wide variants at substantially lower costs than a decade ago [29]. Single-nucleotide polymorphisms (SNPs) are the most common types of genetic variants that are widely distributed throughout the genomes of organisms. They are often associated with disease resistance in plants and virulence in pathogens [25,28,30]. Thus, the use of SNP genotype data has become common in host–pathogen interaction studies. Bi-parental mapping is a widely used technique for mapping traits of interest in plants. This genetic tool has been successfully utilized to characterize loci associated with agronomic, quality, and biotic- and abiotic-stress-related traits. Although time-consuming and resource-intensive, the method is simple and robust. By making appropriate crosses, it can also be used to map loci for desirable traits controlled by rare alleles [31].

William Som (WS, PI 184597), a spring wheat landrace originally from Argentina, was identified to have HTAP resistance in a previous study [32] and has consistently displayed high levels of resistance to stipe rust in our multi-year field evaluations at two different locations, Mount Vernon and Pullman, Washington. The objectives of this study were to (1) identify and map the QTL for HTAP resistance to stripe rust in WS and (2) develop Kompetitive allele-specific PCR (KASP) markers for the identified QTL to be used in marker-assisted selection in breeding programs.

## 2. Results

### 2.1. Stripe Rust Phenotypes

The resistant parent WS (PI 184597) was resistant with infection types (ITs) 2–4 and with a disease severity (DS) range of 2–20%, while AvS was susceptible (ITs 8–9; DS 80–100%) at the adult plant stages (Zadoks GS 60–71) across all seven field environments (Figure 1, Appendix A). The distributions of the stripe rust phenotypes among the recombinant inbred lines (RILs) were skewed towards susceptible reactions, with some variations among environments (Figure 2). The mean IT scores of the RILs ranged from 6.8 to 7.7, and the mean DS values ranged from 58% to 80% across the environments (Appendix A). The analysis of variance (ANOVA) of the stripe rust data (IT and DS) revealed significant (*p* < 2 × 10^−16^) effects of genotype, environment, and genotype × environment interactions for the RIL population (Table 1). The broad-sense heritability values for stripe rust resistance varied from 0.85 to 0.93 across the environments, with the highest heritability obtained with the DS data in the environment in Mount Vernon in 2024 and the lowest heritability measured with the IT data in the environment in Pullman in 2024. Significant (*p* < 0.001) correlations were observed between stripe rust phenotypes recorded in different environments (Figure 3). The Spearman correlation coefficients ranged from 0.47 to 0.82 for IT and from 0.41 to 0.85 for DS between environments. Within an environment, the IT and DS data were highly correlated (0.70–0.96).

### 2.2. Linkage Map

The linkage map of the AvS/PI 184597 RIL population constructed using 1820 SNP markers comprised 19 linkage groups, corresponding to 19 of the 21 wheat chromosomes, with only one marker each for chromosomes 4D and 5D (Appendix A). The genetic map spanned 3067 cM, with an average distance of 0.59 cM between markers. The map lengths of sub-genomes A, B, and D were 1267.10 cM, 1233.24 cM, and 566.82 cM, with a density of 1.43, 1.48, and 5.56 markers per cM, respectively. Among the seven homoeologous wheat groups, the highest number of markers were mapped on group 1 (365 markers) and the lowest number of markers on group 7 (179 markers).

### 2.3. Stripe Rust Resistance QTL

Four QTL were detected for HTAP resistance using the IT and DS data of the RILs evaluated in the seven field environments (Table 2, Figure 4, Appendix A). The resistance alleles of all four QTL were contributed by the resistant parent, WS (PI 184597). These QTL were mapped to chromosome arms 1BL, 2AL, 3AS, and 3BL (Table 2, Figure 4).

The first QTL, *QYrWS.wgp-1BL*, was located on the long arm of chromosome 1B, within the 669 to 682 Mb interval based on the IWGSC RefSeq v2.1, and explained 10% to 19% of the phenotypic variation. The LOD scores for this QTL ranged from 4.5 in Pullman 2023 to 7.4 in Mount Vernon 2024. This QTL was consistently detected across four environments, including Mount Vernon 2015, Mount Vernon 2016, Mount Vernon 2024, and Pullman 2023. The SNP markers flanking this QTL were IWA3998 and IWB7694 to the left (proximal) and IWB40850 and IWB72918 to the right (distal).

The second QTL, *QYrWS.wgp-2AL*, was mapped on the long arm of chromosome 2A at 611–684 Mb and had LOD values ranging from 3.7 to 6.8 across the environments. This QTL was identified in Mount Vernon 2024, Pullman 2016, Pullman 2023, and Pullman 2024. It explained 10.2% to 16.7% of the phenotypic variation. Five SNP markers, IWB7315, IWB57371, IWB51049, IWA5640, and IWB34603, were associated with this QTL.

The third QTL, *QYrWS.wgp-3AS*, mapped to the 9–13 Mb genomic region on the short arm of chromosome 3A, was detected in Mount Vernon in 2015, 2016, and 2024 and in Pullman in 2023. This QTL explained 7.0% to 15.9% of the phenotypic variation, with LOD scores ranging from 3.2 to 6.2 across environments. Three SNP markers tagged the QTL with IWB34643 and IWB63656 to the left (distal) and IWB36767 to the right (proximal).

The fourth QTL, *QYrWS.wgp-3BL*, located on the long arm of chromosome 3B at 476–535 Mb, explained the highest phenotypic variation (12.0% to 27.8%) among the four identified QTL. It was detected in Mount Vernon in 2015 and 2024 and in Pullman in 2016 and 2024. The markers flanking the QTL were IWB11270 and IWA1898 to the left (proximal) and IWB22528 and IWB11836 to the right (distal). The LOD scores for this QTL ranged from 3.5 to 11.5 across the four environments.

### 2.4. KASP Markers

For each of the four QTL, two KASP markers were developed. The sequences of the eight KASP markers developed for the four QTL are provided in Table 3. All eight KASP markers produced the same genotypes for all RILs and the parents as their SNP markers. The results showed that these KASP markers function in the same way as their original SNP markers to distinguish between the SNP genotypes of the four QTL.

### 2.5. Effects of QTL Combinations

The effects of individual QTL and their combinations on stripe rust phenotypes were assessed by grouping the RILs based on their QTL composition (Figure 5A). The mean DS values of the RIL groups carrying any individual QTL, except 3A, were significantly lower (*p* < 0.05) than the group lacking all four resistance QTL. Similarly, all 10 different QTL combinations resulted in statistically significant (*p* < 0.05) reductions in stripe rust severity. Among the two-QTL combinations, 2A + 3B, 1B + 3B, and 1B + 3A were slightly better than 3A + 3B and 2A + 3A. Any combinations of three QTL were almost equally effective. On average, the four-, three-, and two-QTL combinations reduced the stripe rust severity rates by 53%, 45%, and 22%, respectively, compared to lines with no resistance QTL, indicating additive effects of the four QTL.

A similar trend was observed for the IT data, with higher levels of resistance associated with the increase in the number of QTL (Figure 5B). Any individual QTL, except 2A and 3A, and all types of QTL combinations resulted in statistically (*p* < 0.05) lower mean IT scores than the group lacking any of the four resistance QTL.

## 3. Discussion

In this study, we mapped four QTL, *QYrWS.wgp-1BL*, *QYrWS.wgp-2AL*, *QYrWS.wgp-3AS*, and *QYr.WS.wgp-3BL*, for HTAP resistance to stripe rust to the wheat chromosome arms 1BL, 2AL, 3AS, and 3BL, respectively, from the Argentinian wheat landrace WS. These stable QTL had additive effects, with higher resistance observed in RILs carrying more QTL. KASP markers were developed for each QTL, which should facilitate the incorporation of these loci into breeding materials through marker-assisted selection (MAS).

*QYrWS.wgp-1BL* (669–682 Mb) was mapped to the distal region of chromosome arm 1BL. *Yr29* is the only permanently named *Yr* gene conferring HTAP resistance around this position (670–680 Mb) [17,34,35]. As the map positions of *QYrWS.wgp-1BL* and *Yr29* are almost the same and both are associated with HTAP resistance, it is highly likely that *QYrWS.wgp-1BL* is *Yr29*. *Yr29* has been identified in many wheat cultivars and landraces worldwide, suggesting that the gene is very common in wheat germplasm [28,36,37,38]. The prevalence of *Yr29* in wheat varieties may be attributed to selection because of its additional resistance to leaf rust (as *Lr46*) and powdery mildew (as *Pm39*) at the adult plant stages [34,35]. In the present study, the percentages of phenotypic variation explained (PVE) by *QYrWS.wgp-1BL* (*Yr29*) were 10% to 19%. These effects are comparable to the effects of *Yr29* in the ‘AvS/Skiles’ population (PVE = 12–15%) evaluated in Pullman and Mount Vernon, WA in the previous study [28] as in the present study.

*QYrWS.wgp-2AL* was mapped to the distal region of 2AL at 611–684 Mb. Three permanently designated *Yr* genes have been reported on chromosome arm 2AL, including two ASR genes *Yr1* [39] and *Yr32* [40], and one adult plant resistance (APR) gene *Yr86* [41]. Among these, *Yr86* is the closest to *QYrWS.wgp-2AL* but is located ~ 40 Mb distal, at 725–737 Mb, suggesting that it may be a different gene. Within a similar genomic interval as *QYrWS.wgp-2AL*, two HTAP resistance genes/QTL, *Yrxy2* from the Chinese cultivar Xiaoyan 54 [42] and *QYrPI197734.wgp-2A* from the Swedish cultivar Progress (PI 197734) [43], have been mapped to 2AL, and all three likely represent the same resistance locus. Other APR QTL mapped near *QYrWS.wgp-2AL* include *Qyr.gaas.2A.1* [44], *QYr.lrdc-2A.2* [45], and *QYr.Sicau-2AL* [46]. Further studies are needed to determine the relationships of *QYrWS.wgp-2AL* and these QTL.

*QYrWS.wgp-3AS* (9–13 Mb) was discovered on chromosome arm 3AS. To date, no permanently named APR or HTAP resistance genes have been identified on 3AS. However, an ASR gene, *Yr76*, spanning 16–22 Mb [47], resides on this arm in the US club wheat cultivar Tyee. Because of the different resistance types, *QYrWS.wgp-3AS* should be distinct from *Yr76*. Previously, an APR QTL, *QYrto.swust-3AS*, was mapped to 3AS at 7.9–10.21 Mb from the Chinese wheat cv. Toni using the ‘Mingxian 169/Toni’ RIL population [48]. The genomic coordinates of *QYrWS.wgp-3AS* and the previously identified *QYrto.swust-3AS* nearly overlap. However, the latter explained a relatively high phenotypic variation range (31–48%) compared to the 3AS QTL (7–16%). It is not uncommon for the same QTL to exhibit varying effects across genetic studies, as the effects of QTL can be influenced by their genetic background, interactions between QTL, the size of the mapping population, and the mapping method used. Further genetic studies are needed to confirm the relationship between these QTL. Additionally, a marker–trait association (MTA) has also been identified within the *QYrWS.wgp-3AS* confidence interval in a genome-wide association study (GWAS) of Chinese common wheat varieties [49].

On chromosome 3B, five permanently named *Yr* genes have been reported so far. *Yr80* [50], an APR gene, and *Yr82* [51], an ASR gene, are located on the short arm of the chromosome. *Yr80* identified from an Australian wheat landrace Aus27284 was mapped to a 60 Mb interval on 3BL at 565–625 Mb [50]. In the present study, we mapped *QYrWS.wgp-3BL* approximately 30 Mb proximally from the boundary of *Yr80* in the 476–535 Mb region. Further studies are needed to determine whether *QYrWS.wgp-3BL* is the same as or different from *Yr80*. Previously, HTAP resistance QTL *QYrPI197734.wgp-3B* was located on 3B at 423 to 559 Mb [43], which is within the confidence interval of *QYrWS.wgp-3BL*. Interestingly, in the previous study, *QYrPI197734.wgp-3B* explained 13–46% of the phenotypic variation, which is comparable to the PVE values of *QYrWS.wgp-3BL* in the present study. As *QYrWS.wgp-3BL* and *QYrPI197734.wgp-3B* are in the same region and confer HTAP resistance, they should be the same gene. Multiple GWAS studies from different countries have also identified MTAs within the similar region of *QYrWS.wgp-3BL,* further confirming the role of the locus in stripe rust resistance in diverse environments [49,52,53].

The HTAP resistance QTL identified in the present study exhibited additive interaction effects and were highly effective in reducing the stripe rust IT and severity when any three or all four QTL were present together rather than individually in the RILs. This is consistent with the previous findings that most HTAP resistance genes provide only partial or incomplete resistance when present alone [6,25,43]. However, when combined with other HTAP resistance or ASR genes in a gene pyramid scheme, they offer stronger protection, either by enhancing the effects of other genes or through additive interactions [6,25,51]. With the aid of the KASP markers developed in the present study, incorporating three to four of these QTL into lines containing effective ASR genes should be feasible. Combining genes or QTL for HTAP resistance and ASR genes into individual cultivars takes the advantages of both types of resistance. Thus, such a gene pyramid approach ensures that if a predominant race evolves new virulence to overcome one to two resistance genes, the remaining genes can still provide some protection and help reduce crop losses [6]. The wheat breeding program at the International Maize and Wheat Improvement Center (CIMMYT) has successfully combined multiple minor- to intermediate-effect genes to achieve higher levels of stripe rust resistance in wheat cultivars [54]. A similar strategy of integrating HTAP resistance genes with ASR genes in wheat cultivars in the US, especially the PNW region, has proven highly effective in providing durable resistance to stripe rust, preventing major losses in the past 60 years [6,7,25]. The present study demonstrates the effectiveness of the gene combination approach for high levels of durable types of resistance in an Argentinian wheat landrace and provides the information on resistance loci and their KASP markers. Further studies will be conducted to determine the polymorphisms and usefulness of the KASP markers with wheat breeding lines from various regions of the United States.

## 4. Materials and Methods

### 4.1. Plant Materials and Population Development

A cross was made between William Som (WS, PI 184597) and Avocet S (AvS), with AvS used as the female parent. WS is a spring wheat landrace collected from Argentina, deposited in the US National Plant Germplasm System (NPGS) in 1949, and maintained by the USDA-ARS National Small Grains Collection (NSGC) in Abeerdeen, Idaho (https://npgsweb.ars-grin.gov/gringlobal/accessiondetail?id=1159680, accessed on 5 April 2025). We found that WS has effective HTAP resistance and ineffective ASR based on reactions in the fields and seedling and adult plant tests at low and high temperatures in the greenhouse [32]. AvS is an Australian wheat line that is susceptible at all growth stages to most races of *Puccinia striiformis* f. sp. *tritici* (*Pst*) from the US. From the cross, 114 F_5_ to F_8_ RILs were developed using the single-seed descent method in the greenhouse and field.

### 4.2. Stripe Rust Phenotyping

In the present study, the parent F_6_, F_7_, and F_8_ RILs of the AvS/WS were evaluated for their stripe rust responses in seven field environments at two locations, Pullman and Mount Vernon, Washington, under natural infection of *Pst*. These seven environments included field experiments in 2016, 2023, and 2024 in both Pullman and Mount Vernon and 2015 in Mount Vernon. Pullman in eastern Washington and Mount Vernon in western Washington are approximately 500 km apart, separated by the Cascade Range; thus, they have different weather patterns and *Pst* race compositions. Each year, the planting was done between the first and third weeks of April, depending on the weather conditions. Five gram seeds of each line were manually sown in a 50-cm-long row with 20 cm spacing between adjacent rows. The susceptible parent, AvS, was planted after every 20 rows and as spreader rows around the nurseries to increase the stripe rust pressure and promote uniform disease development. The nurseries were arranged in a randomized complete block design with two replications. The IT and DS data were recorded twice, first at the stem elongation stage (GS 31; [55]) and second at the flowering stage (GS 60) in Mount Vernon, and at the flowering stage (GS 60) and the kernel watery ripe stage (GS 71) in Pullman. The IT values were recorded using the 0–9 scale [56], and the DS values were assessed on a 0 to 100% scale based on the modified Cobb’s scale [57].

### 4.3. Stripe Rust Data Analyses

The distributions of the IT and DS data of the RILs across the field environments were visualized using histograms generated with the ggplot2 package in R v4.3.1 [58]. To test the effects of the genotype and environment and their interactions, an ANOVA was performed on the IT and DS data using the ‘aov’ function in R. The broad-sense heritability was calculated using the ‘AOV’ functionality in IciMapping software v4.2 (https://isbreedingen.caas.cn/software/qtllcimapping/294607.htm, accessed on 1 March 2025). The correlation coefficients between the stripe rust data in different environments were computed using the ‘cor’ function, with the Spearman method implemented in the ‘corrplot’ package in R [59].

### 4.4. DNA Extraction, Genotyping, and SNP Calling

For DNA extraction, leaf samples approximately 3 cm in length were collected from the F_7_ RILs, as well as the two parents, at the two-leaf stage grown in the greenhouse. The genomic DNA was extracted using the MagMax^TM^ Plant DNA Isolation Kit (ThermoFisher Scientific, Waltham, MA, USA) on a KingFisher Flex Purification system, following the manufacturer’s instructions. The concentration and purity of the DNA were measured using a NanoDrop 1000 spectrophotometer (ThermoFisher Scientific, Waltham, MA, USA). The RILs and parents were genotyped using the Illumina 90K Infinium iSelect Wheat SNP Array at the USDA-ARS Cereal Crops Genotyping Laboratory, Fargo, ND, USA.

The SNP calling was performed with the software GenomeStudio v2.0 (Illumina, Inc., San Diego, CA, USA). When necessary, alleles were manually assigned to genotype groups (AA, AB, and BB) based on visual assessments of the genotype cluster plots. Monomorphic SNPs and SNPs missing > 10% of data were removed. For the remaining SNP genotypes, imputation was performed using the LD KNNi imputation algorithm implemented in the TASSEL software v5.0 [60]. The chromosomal positions of the SNPs were determined referring the Chinese Spring genome IWGSC RefSeq v2.1 [33].

### 4.5. Linkage Map Construction and QTL Analysis

A Chi-squared goodness-of-fit test was performed to evaluate deviations of SNP genotypes from the expected 1:1 segregation ratio at each SNP locus, using an alpha level of 0.05. The linkage maps were constructed using 1820 high-quality SNPs with MAP functionality in the IciMapping software v4.2 [61]. The SNPs were grouped into linkage groups based on their recombination frequency, using the default threshold value of 0.3. Within each linkage group, the SNPs were ordered using the 2-OptMap algorithm. The SNP order was further refined using the ripple function with a window size of 5, also based on recombination frequency. To associate the SNPs with stripe rust phenotypes, QTL mapping was performed using the Inclusive Composite Interval Mapping (ICIM) method with a step size of 1 cM in IciMapping v4.2, and Kosambi’s mapping function [62] was used to estimate intervals of the stripe rust resistance QTL. A total of 1000 permutations were conducted using a type I error rate of 0.05 to determine the LOD significance threshold for QTL detection.

### 4.6. KASP Marker Development and Genotyping

The KASP markers were designed based on SNPs tagging the QTL. DNA sequences of 150–200 bp in length containing the SNPs were submitted to 3cr Biosciences (Harlow, UK) to design two allele-specific forward primers and one common reverse primer for each SNP marker. The resulting KASP marker sequences were then sent to Millipore Sigma (Sigma-Aldrich, St. Louis, MO, USA) for the oligonucleotide synthesis. Eight KASP markers were developed, with two markers for each of the four QTL.

PCR amplification for KASP genotyping was performed in a 384-well format Biometra TAdvanced thermocycler (Analytik Jena, Jena, Germany) to reduce the reaction volume. Each KASP reaction had a volume of 5 µL per well, comprising 2.5 µL of 2× KASP master mix (3cr Biosciences, Harlow, UK), 0.07 µL of primer mix, 2 µL of DNA at 20 ng/µL, and 0.5 µL of nuclease-free water. The PCR cycling conditions included an initial hot-start activation at 94 °C for 15 min, followed by 12 cycles of touch down PCR (94 °C for 20 s, 68 °C for 1 min, temperature decreasing by 0.9 °C per cycle) and then 35 amplification cycles (94 °C for 20 s, 57 °C for 1 min for annealing and elongation). After PCR amplification, the fluorescence signals were read in a LightCycler 480 II machine (Roche Sequencing and Life Science, Indianapolis, IN, USA), and genotype cluster graphs were generated with the LightCycler 480 software v1.5.1.62.

## Figures and Tables

**Figure 1 ijms-26-05072-f001:**
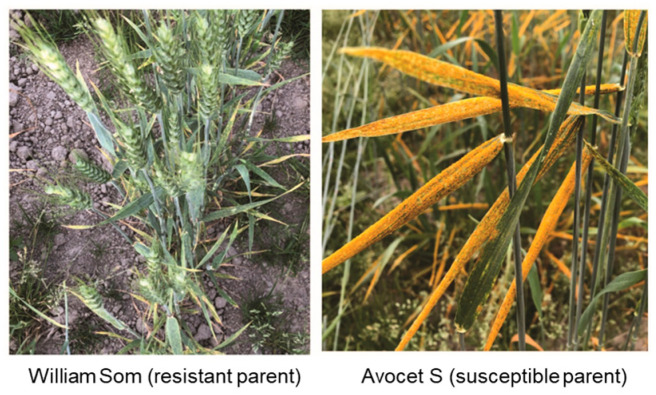
Stripe rust phenotypes on the resistant parent, William Som (PI 184597), and the susceptible parent, Avocet S, observed in the nursery at Mount Vernon, WA, in 2023.

**Figure 2 ijms-26-05072-f002:**
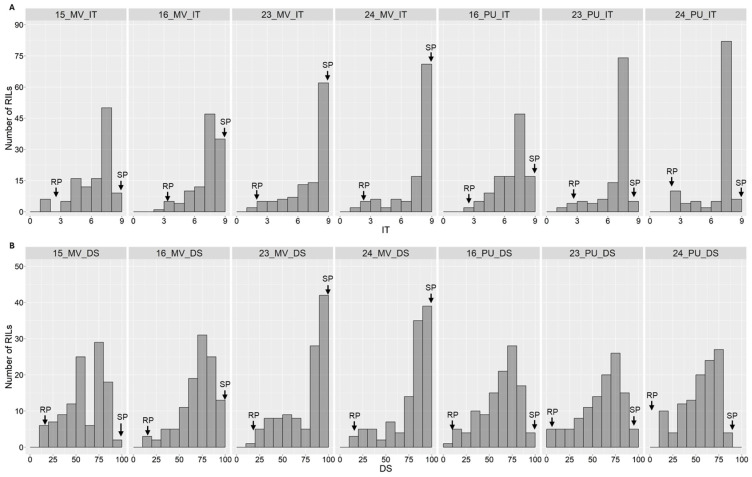
Histograms showing the distributions of stripe rust phenotypes of the recombinant inbred lines (RILs) from cross AvS/WS tested in seven field environments: (**A**) infection type (IT); (**B**) disease severity (DS). The parents and RILs were evaluated for stripe rust responses in Mount Vernon (MV) in 2015, 2016, 2023, and 2024; and in Pullman (PU) in 2016, 2023, and 2024. RP = resistant parent (William Som, PI 184597); SP = susceptible parent (AvS).

**Figure 3 ijms-26-05072-f003:**
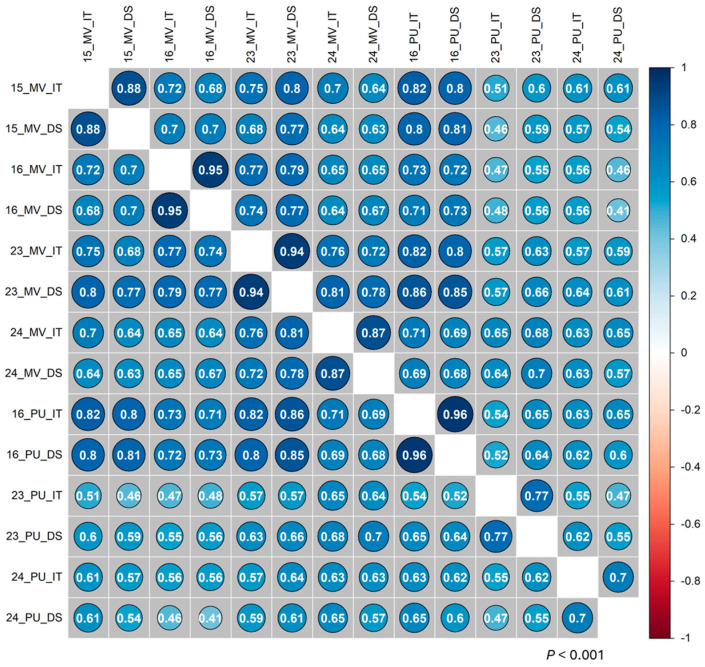
Correlation coefficients between stripe rust phenotypes across seven field environments. The numbers inside the circles are Spearman correlation coefficients, significant at *p* < 0.001; 15 = 2015, 16 = 2016, 23 = 2023, 24 = 2024. MV = Mount Vernon; Pu = Pullman, WA. IT = infection type; DS = disease severity.

**Figure 4 ijms-26-05072-f004:**
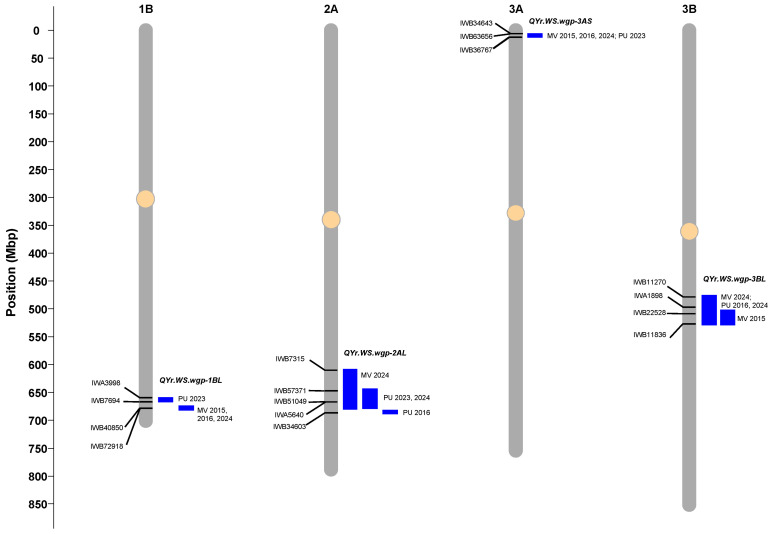
Chromosomal maps of the four quantitative trait loci (QTL) for high-temperature and adult plant resistance identified in this study. QTL names and the environments in which they were identified are shown on the right side of the chromosomes, while markers associated with each QTL are shown on the left. MV = Mount Vernon; PU = Pullman.

**Figure 5 ijms-26-05072-f005:**
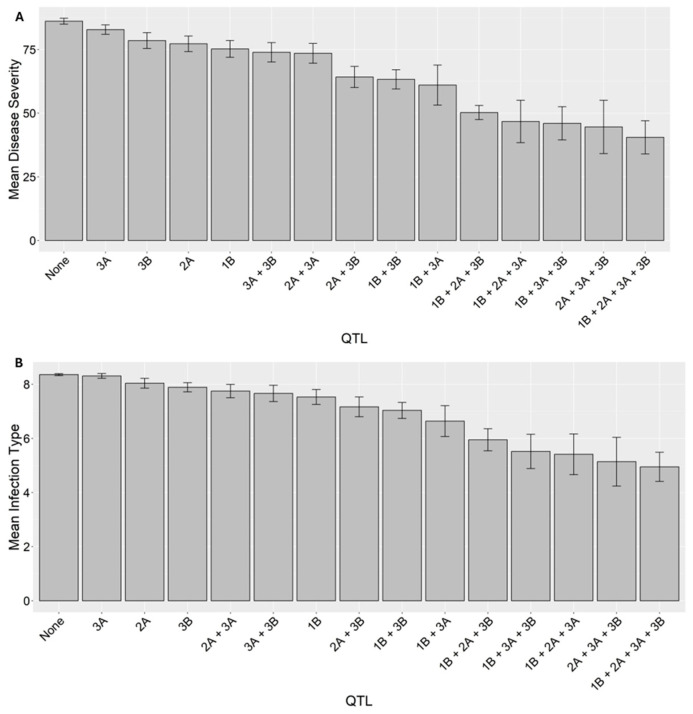
Bar graphs showing the effects of individual quantitative trait loci (QTL) and QTL combinations on the stripe rust disease severity (**A**) and infection type (**B**) of the recombinant inbred lines from the cross AvS/WS.

**Table 1 ijms-26-05072-t001:** An analysis of variance for the infection type and disease severity values of the 114 recombinant inbred lines from the cross AvS/WS tested in seven field environments at Pullman and Mount Vernon, Washington ^a^.

Data	Source of Variations	DF	SS	MS	F	*p* (>F)
Infection type	Genotype (G)	113	3315	29.33	45.83	<2 × 10^−16^
	Environment (E)	6	174	28.98	45.28	<2 × 10^−16^
	G×E	669	1246	1.86	2.9	<2 × 10^−16^
	Error	789	505	0.64		
Disease severity	Genotype (G)	113	546,121	4833	39.86	<2 × 10^−16^
	Environment (E)	6	132,776	22,129	182.52	<2 × 10^−16^
	G×E	669	195,494	292	2.41	<2 × 10^−16^
	Error	789	95,661	121		

^a^ DF = degree of freedom; SS = sum of square; MS = mean square; F = value of estimated variance from means/estimated value from individuals; *p* = probability.

**Table 2 ijms-26-05072-t002:** Four quantitative trait loci (QTL) identified for high-temperature adult plant resistance to stripe rust in spring wheat William Som (PI 184597) from field evaluations.

QTL	Chr.	Interval (Mbp) ^a^	Left Marker	Position (bp)	Right Marker	Position (bp)	LOD ^b^	PVE ^c^	Trait ^d^
*QYrWS.wgp-1BL*	1B	669–682	IWA3998	669,136,631	IWB7694	675,532,521	4.5	13.5	PU_2023_DS
			IWB7694	675,532,521	IWB40850	681,737,056	5.2	15.4	MV_2024_IT
			IWB7694	675,532,521	IWB40850	681,737,056	7.4	19.0	MV_2024_DS
			IWB7694	675,532,521	IWB72918	682,339,545	4.7	10.0	MV_2015_DS
			IWB7694	675,532,521	IWB72918	682,339,545	4.7	12.2	MV_2016_DS
*QYrWS.wgp-2AL*	2A	611–684	IWB7315	611,614,334	IWB57371	643,705,482	3.7	10.2	MV_2024_IT
			IWB57371	643,705,482	IWB51049	678,678,971	4.0	11.0	PU_2023_DS
			IWB57371	643,705,482	IWB51049	678,678,971	5.7	16.7	PU_2024_DS
			IWB57371	643,705,482	IWB51049	678,678,971	4.9	12.0	MV_2024_DS
			IWA5640	679,489,440	IWB34603	684,696,382	6.8	11.9	PU_2016_IT
*QYrWS.wgp-3AS*	3A	9–13	IWB34643	8,940,939	IWB36767	13,212,163	3.5	7.0	PU_2023_DS
			IWB63656	8,942,306	IWB36767	13,212,163	6.2	15.9	MV_2016_DS
			IWB63656	8,942,306	IWB36767	13,212,163	3.2	6.9	MV_2024_DS
			IWB63656	8,942,306	IWB36767	13,212,163	3.3	7.3	MV_2015_DS
*QYrWS.wgp-3BL*	3B	476–535	IWB11270	476,376,513	IWB22528	512,777,195	11.5	27.8	PU_2016_DS
			IWB11270	476,376,513	IWB11836	535,559,634	8.7	15.7	PU_2016_IT
			IWB11270	476,376,513	IWB11836	535,559,634	3.7	12.0	PU_2024_DS
			IWB11270	476,376,513	IWB11836	535,559,634	3.5	12.9	MV_2024_DS
			IWA1898	499,086,813	IWB11836	535,559,634	5.8	12.6	MV_2015_DS

^a^ The physical positions of QTL and markers are based on the IWGSC RefSeq v2.1 genome assembly of wheat cv. Chinese Spring [33]; ^b^ LOD = logarithm of odds; ^c^ PVE refers to phenotypic variance explained by a QTL in percentage. ^d^ The parents and recombinant inbred lines (RIL) populations (F_6_, F_7_, and F_8_) were tested for stripe rust reactions near Mount Vernon and Pullman, WA, in 2016, 2023, and 2024, and additionally in Mount Vernon in 2015. Pullman and Mount Vernon are abbreviated as PU and MV, respectively. IT and DS refer to the infection type and disease severity, respectively.

**Table 3 ijms-26-05072-t003:** Kompetitive allele-specific PCR (KASP) markers developed from the single-nucleotide polymorphism (SNP) markers for the four stable QTL in spring wheat William Som (PI 184597).

KASP Marker	SNP Marker	Primer	Sequence (5′–3′) ^a^	QTL
IWA3998	wsnp_Ex_c4774_8519623	Forward1	GAAGGTGACCAAGTTCATGCTGAGTTTTCAGGCCTTGGAGGG	*QYrWS.wgp-1BL*
		Forward2	GAAGGTCGGAGTCAACGGATTGGAGTTTTCAGGCCTTGGAGGA	
		Common reverse	CTGGGTCGTCAGTTTGACTTAAGCAT	
IWB7694	BS00028747_51	Forward1	GAAGGTGACCAAGTTCATGCTGGACTGGAGCAAAATTTCAAGTGTAA	*QYrWS.wgp-1BL*
		Forward2	GAAGGTCGGAGTCAACGGATTGGACTGGAGCAAAATTTCAAGTGTAG	
		Common reverse	CCCAGCTGCACATTGTAAATTCCGTT	
IWA5640	wsnp_EX_rep_c69799_68761171	Forward1	GAAGGTGACCAAGTTCATGCTAGCCCTTCACCTTGATCACCTT	*QYrWS.wgp-2AL*
		Forward2	GAAGGTCGGAGTCAACGGATTAGCCCTTCACCTTGATCACCTC	
		Common reverse	TCCTTAACGAGGAGCTTGCAGACAT	
IWB34603	IAAV2718	Forward1	GAAGGTGACCAAGTTCATGCTGAAGTTTCAAGATATAAACCAAGTGCATG	*QYrWS.wgp-2AL*
		Forward2	GAAGGTCGGAGTCAACGGATTGAAGTTTCAAGATATAAACCAAGTGCATA	
		Common reverse	CTGCCTAGCCAATCTGTTTATATCTTGTA	
IWB63656	RFL_Contig1488_671	Forward1	GAAGGTGACCAAGTTCATGCTTCCAGTCCAACGCAAGCTGGA	*QYrWS.wgp-3AS*
		Forward2	GAAGGTCGGAGTCAACGGATTCCAGTCCAACGCAAGCTGGG	
		Common reverse	AGGAACAGGCTCAGGGCAGGAT	
IWB36767	Jagger_c8039_67	Forward1	GAAGGTGACCAAGTTCATGCTATGTTAAACATAGGAGTATCACAAAAGATG	*QYrWS.wgp-3AS*
		Forward2	GAAGGTCGGAGTCAACGGATTATAATGTTAAACATAGGAGTATCACAAAAGATA	
		Common reverse	CTTTTGTAGTAACATTTTCTGCTATTGGTA	
IWB11270	BS00082644_51	Forward1	GAAGGTGACCAAGTTCATGCTCAAACCGTATACATGTATGTCTATCCT	*QYrWS.wgp-3BL*
		Forward2	GAAGGTCGGAGTCAACGGATTAAACCGTATACATGTATGTCTATCCC	
		Common reverse	GCTCCGAACCAATCGCCGGTA	
IWB22528	Excalibur_c14999_712	Forward1	GAAGGTGACCAAGTTCATGCTCCTTGTTGATTCTCTCTTCAGAGC	*QYrWS.wgp-3BL*
		Forward2	GAAGGTCGGAGTCAACGGATTCCCTTGTTGATTCTCTCTTCAGAGT	
		Common reverse	CGCTACTCTCCAATGTTTTTGCGAAAATT	

^a^ The allele-specific primers (Forward1 and Forward2) contain 21 bp FAM or HEX tail sequences attached to the 5′ end.

## Data Availability

All data generated in this study are available in the main text and in the Appendix A.

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
