# Peer review of "QTL Mapping and Developing KASP Markers for High-Temperature Adult-Plant Resistance to Stripe Rust in Argentinian Spring Wheat William Som (PI 184597)"

_ijms, 2025, doi:10.3390/ijms26115072_

Round 1
Reviewer 1 Report
Comments and Suggestions for Authors
This manuscript presents the works on QTL mapping and developing KASP markers for the high-temperature adult resistance to strip rust in Argentinian spring wheat William Som, which are important for the proper utilization of this strip rust resitant germplasm in wheat improvement.
The experiment is well designed, and the data is well analysed, the identified QTLs and the developed KASP markers may facilate the use of this resistant germplasm.
There are some concerns and comments as followed:
- Section 2.2, the linkage map was constructed using 1820 SNP markers compsed 19 linkage groups, with 4D and 5D absent? please add the analysis of the genotyping data by 90 K wheat SNP array, and how the linkage map was constructed. In the method, section 4.4, please introduce how the SNP data was linked to the wheat chromosome?
- Section 2.3, Were all the IT and DS data from seven environments used for QTL analysis? Please add the QTLs data for IT and DS, respectively, are there common QTLs for both IT and DS?
- Section 2.4, the primers pf KASP markers should be in the materials and methods, section 4.6.
- In Section 4.5, Chi-squared test was conducted, but no data mentioned.
Author Response
Thanks for reviewing the manuscript. We have revised the manuscript accordingly. Our point-to-point responses are attached in the cover letter.

Reviewer 2 Report
Comments and Suggestions for Authors
QTL Mapping and Developing KASP Markers for High-tem-2 perature Adult-plant Resistance to Stripe Rust in Argentinian 3 Spring Wheat William Som (PI 184597)
This study maps high-temperature adult-plant (HTAP) resistance to stripe rust in an Argentinian spring wheat landrace, William Som (WS). The authors developed a recombinant inbred line (RIL) population from a cross between Avocet S(susceptible) and WS (resistant), and conducted multi-environment field trials. Four quantitative trait loci (QTL)—QYrWS.wgp-1BL, 2AL, 3AS, and 3BL—were consistently identified as contributors to HTAP resistance. Kompetitive allele-specific PCR (KASP) markers were developed for each QTL. The findings offer practical tools for marker-assisted selection in breeding programs aiming for durable resistance to stripe rust.
Comments and Suggestions:
1. The study is scientifically rigorous and well-executed. Identifying four stable QTL and developing reliable KASP markers represent significant contributions to breeding for HTAP resistance in wheat. The inclusion of field evaluations over seven environments enhances robustness and reliability.
2. The multi-environment phenotyping, high heritability estimates (0.85–0.93), and significant genotype-environment interactions are well presented.
3. Statistical analyses including ANOVA and ICIM-based QTL mapping are appropriate. However, including confidence intervals or LOD threshold values in the figures/tables could enhance interpretability.
4. Consider adding more clarity on how KASP marker validation was performed across different populations or breeding lines.
5. Figures are informative, particularly Figure 5 which illustrates the additive effects of QTLs clearly.
6. Ensure figure legends are self-explanatory for a broader audience. For instance, explicitly define all abbreviations (e.g., IT, DS) in each figure legend.
7. The discussion thoroughly compares mapped QTLs to previously identified loci, enhancing the contextual understanding.
8. It might be helpful to speculate on the functional mechanisms or candidate genes underlying these QTLs, particularly QYrWS.wgp-3BL which appears novel.
9. The manuscript is generally well-written with minor grammatical issues. A final proofread could enhance fluency and clarity in some paragraphs (e.g., lines 228–244 and 275–285).
This is a high-quality and impactful study that makes a valuable contribution to wheat genetics and rust resistance breeding. The identification and validation of stable QTLs and associated KASP markers for HTAP resistance in William Som will benefit breeding programs worldwide. I recommend acceptance with minor revisions, mainly for polishing the presentation and enhancing a few interpretative sections.
Comments on the Quality of English LanguageThe manuscript is generally well-written with minor grammatical issues. A final proofread could enhance fluency and clarity in some paragraphs (e.g., lines 228–244 and 275–285).
Author Response
Thanks for reviewing the manuscript. Our point-to-point responses are listed in the cover letter.
